# Large-Scale AI-Based Structure and Activity Prediction Analysis of ShK Domain Peptides from Sea Anemones in the South China Sea

**DOI:** 10.3390/md23020085

**Published:** 2025-02-16

**Authors:** Ziqiang Hua, Limin Lin, Wanting Yang, Linlin Ma, Meiling Huang, Bingmiao Gao

**Affiliations:** 1Engineering Research Center of Tropical Medicine Innovation and Transformation of Ministry of Education, Hainan Key Laboratory for Research and Development of Tropical Herbs, International Joint Research Center of Human-Machine Intelligent Collaborative for Tumor Precision Diagnosis and Treatment of Hainan Province, School of Pharmacy, Hainan Medical University, Haikou 571199, China; huazzq@icloud.com (Z.H.); liminlin239@gmail.com (L.L.); bukaihua991@gmail.com (W.Y.); 2Griffith Institute for Drug Discovery (GRIDD), School of Environment and Science, Griffith University, Nathan, QLD 4111, Australia; linlin.ma@griffith.edu.au

**Keywords:** AlphaFold, sea anemone peptides, Kv1.3 channel, modeling, simulation

## Abstract

Sea anemone peptides represent a valuable class of biomolecules in the marine toxin library due to their various structures and functions. Among these, ShK domain peptides are particularly notable for their selective inhibition of the Kv1.3 channel, holding great potential for applications in immune regulation and the treatment of metabolic disorders. However, these peptides’ structural complexity and diversity have posed challenges for functional prediction. In this study, we compared 36 ShK domain peptides from four species of sea anemone in the South China Sea and explored their binding ability with Kv1.3 channels by combining molecular docking and dynamics simulation studies. Our findings highlight that variations in loop length, residue composition, and charge distribution among ShK domain peptides affect their binding stability and specificity. This work presents an efficient strategy for large-scale peptide structure prediction and activity screening, providing a valuable foundation for future pharmacological research.

## 1. Introduction

Sea anemone peptides are a unique class of toxins derived from sea anemones, recognized as some of the most lethal marine toxins and representing a highly diverse and functionally rich repository of biomolecules [1]. It is estimated that each sea anemone species can produce thousands of peptides, and with over a thousand species formally identified, the total number of sea anemone peptides could potentially reach millions [2]. Despite their distinct structural and functional characteristics compared to other marine toxins, research on their structural characterization and activity prediction remains limited [3]. These peptides typically consist of 27 to 59 amino acids, with molecular weights ranging from 3 to 7 kDa. They can act on a wide array of molecular targets, including voltage-gated sodium (Na_V_) channels, voltage-gated potassium (K_V_) channels, acid-sensing ion channels, and transient receptor potential vanilloid 1 channels [4,5,6,7,8]. Their unique mechanisms of action make sea anemone peptides valuable tools for studying ion channel physiology and pathology while also providing theoretical and practical insights for developing novel analgesics, immunomodulators, and therapeutic strategies for metabolic diseases such as diabetes.

The sea anemone type 1 potassium channel toxin family, also known as Sea anemone toxin k-like (ShK), typically consists of 35–50 amino acid residues and contains highly conserved cysteine patterns forming three disulfide bonds [9]. This specific disulfide bond connection enables these peptides to form a relatively rigid three-dimensional structure, similar to a loop structure, providing crucial support for their stable three-dimensional structure and biological activity. ShK peptides have garnered significant attention due to their high specificity in blocking the Kv1.3 channel, a member of the Kv1 family composed of a tetrameric structure where each subunit contains a pore domain (PD) and a voltage-sensing domain (VSD). The Kv1.3 channel is widely expressed in the immune and nervous systems of mammals, where it plays a pivotal role, particularly in the activation and differentiation of T lymphocytes and various immune cells [10]. Its functional impairment is closely linked to the onset and progression of autoimmune diseases [11]. Unlike traditional immunosuppressive therapies that compromise systemic immunity, increasing the risk of opportunistic infections and tumor formation, ShK peptides can selectively target CCR7-effector memory T lymphocytes by blocking Kv1.3 channels, thereby reducing side effects of systemic immunosuppression and enabling precise immune modulation [12]. Additionally, ShK peptides influence insulin sensitivity and glucose metabolism by reducing inflammatory cytokine production, promoting the glucose transporter type 1 translocation to the cell membrane, and enhancing insulin sensitivity in peripheral tissues [13]. These properties provide a molecular basis for potential interventions for type 2 diabetes and related metabolic disorders.

The ShK derivative ShK-186 (also known as dalazatide) has completed Phase 1b clinical trials demonstrating safety, tolerability and preliminary efficacy in patients with active plaque psoriasis, explaining the therapeutic promise of these peptides [14]. With the rapid development of omics technology, the discovery of sea anemone peptide sequences has increased sharply [15,16]. However, despite their diversity and unique functions, research on their structural characterization and activity prediction remains limited. These peptides, typically 27–59 amino acids long and weighing 3–7 kDa, pose significant challenges for structural analysis using traditional experimental methods like NMR and X-ray crystallography due to their small size, complex disulfide linkages, limited availability, and stringent folding conditions. To date, only a few peptides, such as ShK, have clear structural information, while the vast majority remain unexplored [17]. Given the scale of potential peptide sequences, ranging from tens of thousands to hundreds of thousands, it is impractical to experimentally determine their structure and activity, which makes the conversion of sequence information into structural and functional insights a bottleneck in research.

Advances in computational biology and AlphaFold-driven structure prediction methods have emerged as transformative tools. Traditional protein structure prediction methods include template-based homology modeling and ab initio prediction. Although homology modeling is considered reliable when the similarity with known structures exceeds 25% for novel or distant sequences, homology modeling often fails due to the lack of suitable templates. Ab initio methods based on molecular mechanics and dynamics are limited by the complexity of protein free energy landscapes and algorithmic constraints [18]. Recently, deep learning models like AlphaFold3 have revolutionized the field, integrating extensive evolutionary data, residue pair features, geometric constraints, and protein physicochemical properties to deliver near-atomic resolution predictions directly from amino acid sequences [19]. Despite the remarkable progress AlphaFold has made in enhancing the efficiency and accuracy of peptide structure prediction, it still has limitations in providing dynamic information, simulating real biological environments, predicting protein-ligand interactions, and elucidating functionally relevant mechanisms. To overcome these challenges, this study employs molecular docking and molecular dynamics simulations to evaluate the functional activity of peptides.

This study analyzed ShK domain peptides from four species of sea anemones in the South China Sea, namely *Heteractis magnifica (H. magnifica)*, *Heteractis crispa (H. crispa)*, *Exaiptasia diaphana (E. diaphana)* and *Macrodactyla doreensis (M. doreensis)*. Their amino acid sequences, obtained from our previous transcriptomics studies [20,21,22,23], were analyzed and used to predict 3D structures via AlphaFold, offering insights into the key structural features of ShK domain peptides. Molecular docking studies with the Kv1.3 channel were conducted using these structures, followed by molecular dynamics simulations to optimize the resulting complex models and predict their potential bioactivities. These findings provide a solid foundation for subsequent pharmacological research and highlight the potential of ShK peptides in therapeutic applications.

## 2. Results

### 2.1. Differential Analysis of ShK Domain Peptide Sequences

In the multiple sequence alignment (MSA) of all peptides (Figure 1A), the regions between the six cysteine residues were categorized into Loop1–5, with ShK and HmK as reference sequences [24,25]. The key amino acids within these loops are typically highly conserved [26,27]. Peptides from the *H. magnifica* group generally align more closely with the conserved pattern of the reference peptides, particularly in Loop1 and Loop3, where the key cysteine residues and other structurally critical residues are well-preserved. This suggests that peptides from the *H. magnifica* group maintain a more traditional and stable pattern, with potential functional similarities to the reference peptides. In comparison, peptides from the *H. crispa* group exhibit some degree of diversification, although many structural and functional key residues are retained. Variations are more frequent in peripheral loops, such as Loop4 and Loop5, where changes in charge distribution or hydrophobic residues are observed. These differences in the terminal loop regions may contribute to variations in receptor-binding affinity. Relative to the reference sequence, all peptide groups introduced a higher proportion of acidic amino acids in Loop2 and Loop3, leading to a deviation from the canonical distribution, with the *E. diaphana* group showing the highest proportion. These alterations modify the local charge properties, potentially influencing complex interactions, and introducing novel molecular recognition patterns or regulatory mechanisms.

Compared to other groups, peptides from the *M. doreensis* group exhibit the most distinctive variations, with unique residue substitutions across multiple loops. For example, there is an enrichment of residues like methionine (Met) or tryptophan (Trp) at specific positions. In non-core functional loops such as Loop4 and Loop5, the residue composition diverges significantly from the reference sequences, reflecting unique adaptive evolution. These differences may endow the peptide of the *M. doreensis* group with unique structural flexibility, ligand-binding preferences, or responsiveness to environmental factors.

### 2.2. Typical Structure and Disulfide Bond Connections of ShK Domain Peptides

For the 36 ShK domain peptides without known experimental structures, structural modeling was performed using AlphaFold3. Five models were generated for each peptide, with the highest-scoring model selected for detailed analysis. Figure 2 displays the three-dimensional structures of the ShK domain peptides, along with their scores and quality assessments. Among the generated models, 30 successfully exhibited the characteristic ShK domain geometry and correct disulfide bond connections, with predicted TM-score (pTM) values exceeding 0.5. The remaining six models failed (pTM < 0.5)—most of which belonged to the *M. doreensis* group, likely due to sequence mutations introducing larger residues that disrupted spatial configuration and led to structural disorder. In the successful models, the typical features of the ShK domain were clearly visible: the core region, composed of residues between the first and sixth cysteines, was prominently displayed. Most models showed three α-helices, while a few had one α-helix partially replaced by turns and random coils, though retaining some helical characteristics. Notably, in all models, the terminal regions (before Cys1 and after Cys6) predominantly displayed random coil configurations in the predicted structures, in contrast to the stabilized Loop1–5 regions supported by disulfide bonds. In the scoring visualization, these terminal regions were represented with lighter colors compared to the core regions, and their higher Expected Position Error (EPE) values indicated greater structural flexibility (Figure 2).

### 2.3. Comparative Analysis of Loop Flexibility in ShK Domain Peptides

The flexibility of loops in ShK domain peptides varies significantly and can be ranked based on their structural rigidity and variability. Loop4, consisting of only three amino acid residues, exhibits the highest rigidity, as reflected in its lowest EPE score and minimal structural variation (Figure 2). Loop1, which contains 5–9 residues, also demonstrates relatively high rigidity and conservation, although it is slightly less rigid than Loop4. In contrast, Loop2 (4–13 residues) and Loop3 (4–15 residues) are the most flexible, with the highest and second-highest EPE scores, respectively, indicating greater structural variability and potential for adaptation, which could contribute to the formation of new active sites. Loop5, despite being only two residues long, shows weaker rigidity due to the presence of the highly conserved glycine (Gly) at the first position. Gly’s minimal steric hindrance facilitates turn formation and hinders stable α-helical structures.

The differences in flexibility among these loops reflect their roles in structural stability, functional specificity, and the potential for novel interactions within the ShK domain peptides. Notably, the EPE score ranking aligns with sequence similarity results, where lower-scoring loops show higher sequence similarity and greater conservation. Additionally, in the Loop3 domain, peptides such as HM-55/56, ED-88, and MD-384 have a Loop3 length of 14 to 15 residues, characterized by an abundance of seven to eight lysine (Lys) and arginine (Arg) residues with basic side chains. These basic amino acids contribute to distinct properties: Lys, with its positive charges and a long side chain, facilitates the formation of amide bonds with carboxyl-containing small molecules or proteins, enabling stable non-bond interactions. Arg, on the other hand, enhances proper protein folding and stability through ionic bonds, hydrogen bonds, and cation–π interactions [28,29]. These combined factors significantly increase the likelihood of active sites being present in the Loop3 region.

### 2.4. Analysis of the ShK-Loop

The ShK domain peptides are anchored by disulfide bonds at three critical positions [25], forming a cross-linked cyclic topology, with C1–C6 forming a large outer loop, C2–C4 forming a moderately sized inner loop, and C3–C5 stabilizing the core structure (Figure 1B). These disulfide bonds are localized on the same side and form two internal cross-linkings, resulting in the unique “ShK-Loop” conformation. This distinctive topology enables these peptides to precisely match the Kv1 channel in a “key-lock” binding pattern, ensuring selective Kv1 channel inhibition without significantly affecting other potassium channels.

Classical ShK domain peptides typically possess 2–3 α-helices, which constitute their essential secondary structural core (Figure 2). These helices are located across the N-terminal, central, and C-terminal regions, with adjacent α-helices arranged in a parallel or perpendicular orientation. However, deviations may occur when multiple glycine (Gly) and proline (Pro) residues accumulate near the C-terminal, potentially disrupting the regular α-helical structure and promoting locally disordered or flexible regions [30]. This subtle structural plasticity allows certain peptides to deviate from the reference model while retaining their core functional motifs.

### 2.5. Molecular Docking Between ShK Domain Peptides and the Kv1.3 Channel

This study utilized AlphaFold3 predicted models and reference models for large-scale molecular docking with the Kv1.3 channel (Figure 3). The results demonstrated that all correctly predicted structures and reference models successfully docked with the Kv1.3 channel, and they could be categorized into two distinct interaction modes. Peptides from the *H. magnifica*, *H. crista*, and *E. diaphana* groups primarily interact with the PD of the Kv1.3 channel, classifying them as pore-blocking toxins. In contrast, peptides from the *M. doreensis* group displayed unique characteristics, deviating from the traditional ShK domain peptides by mainly interacting with the VSD of the Kv1.3 channel, suggesting their role as gating-modifier toxins. Among the four groups of peptides that successfully docked, the *E. diaphana* group showed the best docking performance, with its optimal peptide, ED-88, being the most effective overall. This was followed by the *H. magnifica* group, where the optimal peptide HM-53 ranked second overall. In comparison, peptides from the *M. doreensis* and *H. crista* groups exhibited lower performance, with their best peptides, MD-385 and HC-46, ranking ninth and tenth, respectively. Among the reference peptides, HmK ranked fifth, while ShK tied with HC-46 for the tenth position (Figure 3A).

#### 2.5.1. Salt Bridge Analysis

Salt bridges are essential non-bond interactions in protein structures, combining hydrogen bonding and electrostatic forces [31]. They are formed between two oppositely charged amino acid residues within a proximity of less than 4 Å and play a critical role in structural stability and specific molecular recognition. Most ShK domain peptides retain Lys or Arg in Loop3 or Loop2 as key residues [2,16], which frequently interact with negatively charged glutamic acid (Glu) or aspartic acid (Asp) residues in the PD of the Kv1.3 channel to establish salt bridges. For instance, ED-88 often forms salt bridges between Lys in Loop3 and Glu residues in the channel protein. This interaction is central to the mechanism by which ShK domain peptides block the Kv1.3 channel, involving direct interaction with the negatively charged region of the channel [32]. The side chain of Lys directly inserts into the pore region, thereby obstructing potassium ion flow. Compared to the reference peptides, the *E. diaphana* group exhibits greater diversity in salt bridge interactions, although the overall regions where these interactions occur remain largely consistent. This is attributed to the fact that the *E. diafana* peptides are rich in acidic residues within Loop2 and Loop3, which interact with positively charged residues in the PD of the Kv1.3 channel to form reverse salt bridges, further blocking potassium ion flow. In contrast, the *M. doreensis* group demonstrates a more random distribution of salt bridges, with occasional occurrences in Loop1, Loop4, and Loop5. This suggests that *M. doreensis* peptides may possess broader binding adaptability compared to peptides from other groups. The peptides from the *H. magnifica* and *H. crispa* groups exhibit similar behavior to the reference peptides (Figure 3B,C).

Overall, this study found that salt bridges are highly concentrated in the Loop3 region of ShK domain peptides, which is typically a functional hotspot responsible for binding to the Kv1.3 channel. Predictions indicate that ED-88 and HmK can form up to five salt bridges with the Kv1.3 channel. Within the Kv1.3 channel, salt bridges are predominantly located in two regions: near R350 in the VSD and near G448 in the PD. Most residues involved in these interactions are Glu, acting as H-Acceptors (negative), with a smaller number of Arg residues functioning as H-Donors (positive). For ShK domain peptides, salt bridge formation is primarily localized to Loop3 (with a smaller contribution from Loop2), typically involving Lys and Arg as H-Donors (positive). Contributions from other loop regions are rare. Interestingly, despite the relatively stable structure of Loop4 and its conserved aliphatic residues at the first two positions, no salt bridges are formed in this region (Figure 3). This is likely due to its position at the C-terminal, oriented outward from the complex, resulting in spatial distances too great for effective interaction with nearby negatively charged residues.

#### 2.5.2. Electrostatic and Hydrogen Bond Analysis

Electrostatic interactions, which arise from the long-range attraction between oppositely charged groups, play a crucial role in enhancing binding stability [33]. Across all groups of ShK domain peptides, Loop3 consistently exhibits strong electrostatic attraction. This interaction is primarily driven by positively charged residues in the peptides binding to negatively charged regions on the Kv1.3 channel surface. The classical Arg residues in the reference peptides are well preserved in the peptides of *H. magnifica* and *H. crispa* groups, enhancing electrostatic interactions. Peptides from the *E. diaphana* group exhibit greater diversity in electrostatic interactions due to the abundance of negatively charged residues in the Loop2 and Loop3 regions. These residues enhance the ability of the peptides to interact with positively charged regions on the Kv1.3 channel. In contrast, peptides from the *M. doreensis* group exhibit variability in the distribution of electrostatic interactions, likely influenced by non-conserved sites.

Predicted models indicate that ED-88 can form up to eight electrostatic interactions with the Kv1.3 channel, primarily involving Loop2 and Loop3 regions. These regions interact significantly with positively (or negatively) charged regions on the channel, making ED-88 the top performer among all peptide groups (Figure 3). Similar to salt bridges, the regions involved in electrostatic interactions partially overlap with salt bridge formation sites but involve a greater number of residues. Additionally, the central region of Loop3, containing conserved aromatic residues, contributes unique π interactions within the electrostatic binding, further stabilizing the complex.

Hydrogen bonds, a special form of electrostatic interaction, are critical for the stability and structural integrity of protein complexes [34]. Peptides across all groups exhibit some level of conservation in hydrogen bond formation, primarily involving Lys and Arg residues in Loop3 binding to the Kv1.3 channel. Compared to other groups, peptides from the *E. diaphana* group form relatively more hydrogen bonds, as their residues are more inclined to establish stable polar interactions with positively charged regions on the Kv1.3 channel. In contrast, hydrogen bond distribution in peptides from the *M. doreensis* group is more dispersed, with some atypical hydrogen bond sites observed in Loop1, Loop4, and Loop5.

Predictions indicate that ED-88 can form up to 10 hydrogen bonds with the Kv1.3 channel, ranking highest among all peptides (Figure 3). These bonds are primarily concentrated around the G448 region of the channel (Selective Filter and Outer Vestibule segments in the PD), overlapping with salt bridge formation sites but involving a broader range of residues. In ShK domain peptides, hydrogen bond-forming residues are predominantly located in Loop3 and Loop2, but residues in Loop1 and Loop4/5 contribute more significantly than in salt bridge formation (Figure 3B). This is mainly due to the participation of small, sulfur-free residues such as serine (Ser) and threonine (Thr). Ser, with its highly polar hydroxyl group, is particularly conducive to hydrogen bond formation, while Thr, with its chiral carbon in the side chain, provides two distinct conformations, enabling versatile participation in protein folding and interactions.

#### 2.5.3. Hydrophobic and Adverse Interaction Analysis

Hydrophobic interactions, arising from the stable hydrogen-bond network among water molecules, cause hydrophobic molecules to cluster together to minimize contact with water [35]. Compared to other interactions and hydrogen bonds, hydrophobic interactions are generally weaker and lack directionality, as hydrophobic molecules can aggregate in any orientation. However, when multiple hydrophobic interactions accumulate over a larger contact area, they can significantly enhance molecular affinity and binding stability [36]. All ShK domain peptides from different species of sea anemones exhibit strong hydrophobic interaction tendencies in the Loop3 and Loop4 regions. Aromatic residues in Loop3, such as phenylalanine (Phe) and tyrosine (Tyr), often interact with the hydrophobic pockets of the Kv1.3 channel, stabilizing the peptide-channel complex [32]. Core hydrophobic residues from the reference peptides, such as Phe, are well conserved in the *H. magnifica* and *H. crispa* groups, contributing to their structural stability. Notably, peptides from the *M. doreensis* group exhibit significant differences in hydrophobic interaction patterns, with Loop1–5 containing more aromatic residues. These residues may engage in non-specific interactions with the hydrophobic surfaces of Kv1.3. Among these ShK domain peptides, MD-385 is predicted to form the highest number of hydrophobic interactions (10), while peptides from the *M. doreensis* group exhibit the highest average number of hydrophobic interactions (6.8) (Figure 3B). The core region of hydrophobic interactions involves the hydrophobic zones of the Kv1.3 channel interlocking with aromatic amino acids on Loop3 of the peptide to stabilize the interaction between the two.

Adverse interactions, primarily including unfavorable hydrophobic contacts and repulsive electrostatic interactions, negatively impact binding affinity [37]. Unfavorable hydrophobic contacts, often caused by steric hindrance, are primarily observed between Loop1/2 residues of the peptides and the Kv1.3 channel. Repulsive electrostatic interactions mainly occur between positively charged Arg residues in the Kv1.3 channel and Arg or Lys residues in Loop2/3 of the peptides (Figure 3C). These adverse interactions increase binding energy, thereby reducing the affinity between the ligand and the receptor. Peptides from the *M. doreensis* group exhibit higher levels of unfavorable hydrophobic contacts, largely due to their low sequence conservation. This is particularly evident in Loop4/5, where the introduction of bulky residues such as Trp and Tyr increases steric hindrance and repulsive effects (Figure 1A).

### 2.6. Molecular Dynamics Simulation of the Complex Docked by ShK Domain Peptides and Kv1.3 Channel

To gain deeper insights into the binding process between ShK domain peptides and the Kv1.3 channel, the top-ranked ShK domain peptides from each group, along with reference peptides ShK and HmK, were selected for molecular dynamics simulations. Complexes were constructed between these peptides and the Kv1.3 channel protein, and simulations were performed using the Chemistry at Harvard Macromolecular Mechanics (CHARMm) force field within a solvated water model with defined boundaries to closely mimic physiological conditions. For example, in the ED-88 and Kv1.3 channel complex, the system was solvated with 29,721 water molecules, 78 sodium ions, and 78 chloride ions to ensure a realistic simulation setup (Figure 4).

#### 2.6.1. RMSD and RMSF Analysis in Simulations

Root mean square deviation (RMSD) is a key metric for evaluating the deviation of a system’s conformation from its reference structure during simulations, reflecting the structural stability of the complexes [38]. During the 50 ns molecular dynamics simulation (Figure 4B), the RMSD values for all six complexes were relatively high in the initial phase, primarily due to strong interactions between the Kv1.3 channel protein and surrounding solvent molecules, causing substantial structural fluctuations in water-exposed amino acid residues. However, as the simulations progressed, most complexes stabilized, with RMSD values falling below 1 after 40 ns. This indicates that these complexes achieved relatively stable conformations in the later simulation stages. Based on the average RMSD at 50 ns, the complex formed by ShK and the Kv1.3 channel exhibited the best performance with a value of 1.0797 nm, indicating excellent stability. However, further analysis of the average RMSD during the last 10 ns, when these complexes reached relative stability, showed that ED-88 outperformed ShK, showing the lowest average RMSD value (0.6614nm), while ShK ranked second (0.6772 nm).

Root mean square fluctuation (RMSF) is a key metric used to evaluate the structural fluctuations of individual amino acid residues in a protein during simulations, reflecting the flexibility of molecular fragments. Figure 4B illustrates the RMSF distributions of both the peptide and the Kv1.3 channel when forming complexes. In the complexes, the Kv1.3 channel exhibited significant fluctuations near residues R350 and M450, corresponding to the VSD and PD, respectively. Additionally, near residue A265, located at the periphery of the Kv1.3 channel, lies an unstructured coil without a defined secondary structure, leading to significant structural fluctuations and a correspondingly high RMSF value. Overall, the Kv1.3 channel residues in the VSD exhibited a lower average RMSF value (0.4876 nm) when bound to MD-385, whereas the PD had the highest average RMSF value (0.6351 nm). This indicates that MD-385 primarily acts on the VSD of the Kv1.3 channel, reducing residue fluctuations in this area. In contrast, the average RMSF values in the PD were lower than those observed with MD-385 for the other five peptides: 0.4841 nm (ED-88), 0.4998 nm (ShK), 0.5197 nm (HC-46), 0.5531 nm (HmK), and 0.5676 nm (HM-53). This suggests that these peptides primarily target the PD, blocking it and reducing residue fluctuations in this region. After binding to the Kv1.3 channel, most ShK domain peptides showed a trend of higher RMSF values at the termini and lower values in the middle. MD-385, however, exhibited higher fluctuations not only at the termini but also in Loop2, resulting in the highest average RMSF value of 0.5398 nm, indicating its lower structural stability compared to other peptides.

#### 2.6.2. Binding Free Energy and Non-Bond Dynamics Analysis in Simulations

Binding free energy serves as an indicator of the interaction strength between peptides and proteins in a complex, reflecting the cumulative effect of thousands of individual interactions [39]. These interactions can be classified as favorable or unfavorable. When the total energy of favorable interactions exceeds the energy of unfavorable ones, the binding process occurs spontaneously. Figure 4B shows the changes in binding free energy of the six complexes during the 50 ns simulation. All complexes consistently exhibited negative binding free energy values throughout the simulation, with fluctuations remaining within a reasonable range (±1%), indicating sustained binding strength and structural stability. The complex formed by ED-88 and the Kv1.3 channel had the lowest average binding free energy (−256302.32 kcal/mol), followed by HC-46 (−212753.84 kcal/mol) and ShK (−209820.1 kcal/mol), with HC-53 having the highest average binding free energy (−172233.7 kcal/mol). This suggests that ED-88 exerts the strongest blocking effect on the Kv1.3 channel among all the peptides studied.

The complexes formed between peptides from the *H. magnifica*, *H. crista*, and *E. diafana* groups and the Kv1.3 channel exhibited stable non-bond interactions, primarily between the Loop3 of peptide and the PD of the Kv1.3 channel. In contrast, the peptide from the *M. doreensis* formed non-covalent interactions between all Loops and the VSD of the Kv1.3 channel. Predictions indicate that the complex formed by ED-88 and the Kv1.3 channel may involve up to two salt bridges, seven electrostatic forces, 29 hydrogen bonds, and 13 hydrophobic interactions. Of these, one salt bridge, three electrostatic forces, 18 hydrogen bonds, and four hydrophobic interactions demonstrated high stability and persistence (formation probability > 60%) (Figure 5). This represents the best performance among all peptides. Importantly, these stable non-bond interactions include those formed between the Arg-Tyr functional dyad in Loop3 of the peptide and residues in the PD of the Kv1.3 channel. These data strongly support the potent blocking effect of ED-88 on the Kv1.3 channel, surpassing the reference peptides ShK and HmK.

## 3. Discussion

In recent years, the application of AI in the biomedical field has achieved remarkable progress, opening up new horizons for peptide research [19]. Sea anemone peptides are a class of bioactive peptides characterized by diverse functions, remarkable sequence diversity, and complexity. They feature a rich variety of amino acids and exhibit both conservation (with highly conserved cysteine residues) and variability (in the length and amino acid composition of Loop1–5) [2,3,9,16]. With the rapid development of multi-omics technologies, thousands of peptide sequences have been identified from sea anemones [40,41]. However, the vast volume of these peptide sequences makes it challenging to conduct structural analysis, target screening, and activity validation individually using traditional experimental methods, as these processes are time-consuming, labor-intensive, and costly. Against this backdrop, the integration of AI offers an innovative solution to these challenges. By combining high-confidence AI predictions, researchers can identify highly active candidate molecules from a large pool of peptide sequences, followed by experimental validation of their structure, targets, and functional activities. The complementary use of AI predictions and experimental validation not only significantly improves research efficiency but also substantially reduces experimental costs, laying a solid foundation for achieving scalable and efficient peptide research.

In this study, sequence analyses were performed on four groups of peptides from South China Sea anemones (*H. magnifica, H. crispa, E. diaphana*, and *M. doreensis*), using ShK and HmK as reference sequences. The results indicate that these peptides exhibit distinct evolutionary patterns. Overall, the *H. magnifica* group remains closely aligned with the canonical distribution patterns observed in established ShK domain peptides. Here, residue spacing and side-chain chemistry are both well-conserved, suggesting a maintained functional output that parallels the activity and specificity of the reference sequences. In parallel, the *H. crispa* group, though somewhat more flexible in its sequence composition, still retains a substantial structural and functional resemblance to ShK. Both of these groups share high similarity in their loop regions, including minimal alterations in loop length and the preservation of key residues pivotal for charge interactions, aromatic stacking, and target channel recognition (Figure 1A). Such conserved features indicate that *H. magnifica* and *H. crispa* peptides may operate under evolutionary pressures favoring stability and high-affinity interactions.

In contrast, the other two groups, *E. diaphana* and *M. doreensis*, deviate from classical ShK structures. The *E. diaphana* group introduces acidic residues (e.g., Glu, Asp) into multiple loops, thereby shifting and altering the character of complex interfaces. This modification may enhance or diversify binding properties, allowing these peptides to recognize different receptor variants or adapt to changing ecological pressures [42]. The *M. doreensis* group, on the other hand, exhibits even more extensive adaptive modifications. The increased proportion of bulky or hydrophobic residues in Loop1–5 imposes higher steric hindrance and can disrupt the formation of a perfectly “ring-like” ShK-Loop structure. Such structural deviations can profoundly influence loop flexibility, overall folding stability, and the peptide’s spatial orientation toward its molecular targets [43]. Consequently, the *M. doreensis* group peptides may adopt more dispersed and less stable conformations in predictive models, highlighting the crucial role of sequence composition in shaping and fine-tuning the functional topology and specificity of ShK domain peptides. Notably, the majority of peptides retained the critical Lys-Phe or Tyr or Trp functional dyad in Loop3 (Figure 5B), which represents the pivotal region responsible for the highly specific blocking effect of ShK domain peptides on the Kv1.3 channel [32,44].

ShK domain peptides form a cross-linked ring topology through disulfide bonds: C1-C6 creates a large outer ring, C2-C4 forms a medium-sized inner ring, and C3-C5 stabilizes the core structure (Figure 2). This arrangement results in a more symmetric topology but lacks the interlocking “knot-like” structure characteristic of the inhibitor cystine knot (ICK) motif (Figure 1B). The fundamental difference lies in the disulfide bond connectivity; ICK-like peptides are connected by C1-C4, C2-C5, and C3-C6 [45]. The knot-like topology of ICK-like peptides typically compacts the peptide into a near-spherical structure dominated by β-sheets. In contrast, the disulfide bonds in ShK domain peptides are concentrated on the same side, with two disulfide bonds cross-linking internally, resulting in an elongated and twisted ring topology that visually resembles a distorted α-helix. We refer to this unique conformation as the ShK-Loop. In terms of stability, the ICK-like peptides are more robust due to their interlocking disulfide topology, which locks the overall structure and enhances both stability and functional specificity [46]. This is further supported by its β-sheet-dominated secondary structure. In comparison, the ShK-Loop exhibits slightly lower stability, as its cross-linked ring topology allows greater flexibility. The ShK-Loop is primarily composed of α-helices stabilized by disulfide bonds in the core region, while its peripheral loop regions display structural diversity, enabling it to adapt to various molecular recognition tasks. This flexibility makes the ShK-Loop particularly suited for dynamic binding and versatile functions, such as specific interactions with receptors or ion channels [47,48,49].

Molecular docking studies revealed distinct binding models among the four sea anemone peptide groups. Peptides from the *H. magnifica* and *H. crista* groups primarily target the PD of the Kv1.3 channel as pore-blocking toxins [50]. Their binding mode is balanced, involving salt bridges, electrostatic interactions, and hydrogen bonds, with high affinity and binding specificity, similar to reference peptides. *E. diaphana* peptides also target the PD but emphasize charge-driven interactions, leveraging positively and negatively charged amino acids to enhance electrostatic binding and adaptability to specific Kv1.3 channel mutants or variant subtypes.

In contrast, the *M. doreensis* group displays unique behavior, primarily acting on the VSD in the Kv1.3 channel as gating-modifier toxins [51]. This group is characterized by an increased number of hydrophobic residues (e.g., Met, Leu, Ile, Val) and bulky residues (e.g., Trp, Phe) while losing major positively charged residues. This shift results in prominent hydrophobic interactions but reduced contributions from salt bridges and electrostatic interactions. These changes may confer broader adaptability at the cost of reduced specificity, particularly on the hydrophobic surface regions of the Kv1.3 channel, where non-specific binding often dominates. Additionally, the introduction of bulky residues may cause spatial exclusion, unfavoring superficial binding to the Kv1.3 channel. Compared to other peptide groups, *M. doreensis* group peptides primarily rely on hydrophobic interactions and partial hydrogen bonding. While this binding mode lacks specificity, it provides advantages in broad-spectrum binding and functional diversity. It is hypothesized that peptides from the *M. doreensis* group may function similarly to Hanatoxin, as they also primarily interact with the VSD of the Kv1.3 channel [52]. These peptides typically exhibit inhibitory effects on multiple Kv channels with low specificity, a property that markedly distinguishes them from traditional ShK domain peptides.

This study further utilized molecular dynamics simulations to investigate the interaction dynamics and stability profiles of ShK domain peptides. Significant RMSD deviations were observed in the early stages of the simulation (Figure 4B), primarily due to solvent effects and conformational redistribution [53]. Over time, as the system approached equilibrium, the RMSD values of all complexes decreased and stabilized, indicating a gradual transition from a disordered to an ordered state. This suggests that, despite differences in sequence and docking modes, the peptides promote the formation of stable binding states with the channel protein in solution. Among the complexes, ED-88 exhibited the greatest RMSD reduction toward the end of the simulation, indicating its distinct ability to achieve conformational locking, which positions it as a promising candidate for a high-stability Kv1.3 inhibitor. The RMSF results provide a more detailed, residue-level perspective on the conformational fluctuation characteristics of the system. Specific regions of the Kv1.3 channel, such as those near Ile370 and Gly448, corresponding to the core sites for pore-blocking toxins and gating modifiers, respectively, exhibited relatively high flexibility during the simulation. These regions align with naturally occurring conformational “hot spots” or dynamic key sites within the channel [54]. Upon binding with these peptides, the Kv1.3 channel exhibits a reduction in local fluctuations, suggesting that these peptides may possess a unique ability to regulate local channel dynamics (Figure 4B). Moreover, during the simulation process, ED-88 exhibited the lowest average binding free energy, indicating a significant binding advantage under the conditions of this study. This may be attributed to the presence of two functional dyads, Arg-Tyr and Lys-Phe, in Loop3 of ED-88, which form stable and persistent non-bond interactions with the Kv1.3 channel [55].

Despite the revolutionary advances in peptide structure prediction achieved by AlphaFold3, several limitations remain, including the lack of dynamic information, deviations from real biological environments (such as the absence of solvents and ions during prediction), and limited capabilities in predicting ligand-protein interactions. To address these shortcomings, this study integrates molecular docking (to predict ligand-protein interactions) and molecular dynamics simulations (to model realistic biological environments and dynamic properties). Using this approach, the study resolved structural and functional relationships of ShK domain peptides and their binding mechanisms with the Kv1.3 channel. Furthermore, this method can facilitate the rapid identification of candidate peptides with high affinity and specificity, paving the way for experimental validation and therapeutic exploration.

## 4. Materials and Methods

### 4.1. Alignment

A total of 36 ShK domain peptide sequences were identified from four sea anemone species (*H. crispa*, *E. diaphana*, *H. magnifica*, and *M. doreensis*) collected from the South China Sea. These sequences were supplemented with two well-characterized reference peptides, ShK and HmK, to provide a comprehensive dataset for comparative analysis [56].

All sequences were curated and saved in FASTA format. Sequence Alignment: The FASTA-formatted sequences were imported into SnapGene Viewer, where MUSCLE (Multiple Sequence Comparison by Log-Expectation) was employed for multiple sequence alignment. The alignment was conducted using default parameters, ensuring consistency across sequences. To achieve optimal alignment of functionally critical residues, including cysteine residues involved in disulfide bonding, manual adjustments were applied where necessary. The aligned sequences were visualized with amino acid residues colored according to their physicochemical properties using the Zappo color scheme for enhanced interpretability. Conservation analysis was conducted to assess sequence conservation across the aligned ShK peptides; the alignment results were exported and processed using the WebLogo tool (http://weblogo.berkeley.edu/, accessed on 20 November 2024). This tool was used to generate a sequence conservation logo that visually represents the degree of conservation at each position within the alignment. Residue heights in the logo reflect their relative frequency, providing insights into conserved functional and structural motifs, particularly within the ShK domain.

### 4.2. Phylogenetic Analyses

The construction of the phylogenetic tree was performed based on multiple sequence alignment results generated using SnapGene Viewer 8.0. Initially, the sequences were aligned using SnapGene’s built-in multiple alignment algorithm. The alignment results were carefully inspected for quality and consistency. Non-alignable terminal sequences were identified and removed using MEGA 11 to ensure that all sequences maintained complete alignment without introducing gaps or ambiguities that could affect downstream analysis [57].

The phylogenetic tree was constructed using the Maximum Likelihood method, implemented within the MEGA 11 software package. The following parameters were applied during the analysis. Model selection: The substitution model for ML analysis was automatically selected based on the best-fit model calculated using the Akaike information criterion within MEGA 11. Bootstrap analysis: To assess the statistical reliability of the tree topology, a bootstrap resampling approach with 2000 replicates was employed. This robust resampling provided a measure of confidence for each branching node within the tree. Branch lengths were estimated based on the number of substitutions per site. Other parameters, and all other settings were left at their default values, ensuring a standard and reproducible analysis.

The resulting phylogenetic tree was visualized and annotated within the MEGA 11 platform, allowing for the interpretation of evolutionary relationships among the analyzed sequences. Nodes with bootstrap values below 50% were considered unreliable and were subsequently marked or excluded from the analysis.

### 4.3. AlphaFold3 Modeling

Protein structure prediction: The three-dimensional structures of the target proteins were predicted using AlphaFold v3.0 (https://alphafoldserver.com/, accessed on 28 October 2024) [19]. Amino acid sequences of the target proteins were submitted to the AlphaFold server, which integrates advanced machine learning techniques with multiple sequence alignments and structural template information to generate high-accuracy protein models.

Model selection: Among the five structural models generated by AlphaFold for each protein, the top-ranked model was selected based on the pTM score. This score provides a global measure of model quality, with higher values indicating greater confidence in the overall structural prediction. 

Residue-specific quality evaluation: To assess the quality of individual residues within the predicted structures, the EPE metric was analyzed. Residues with lower EPE values were interpreted as being predicted with higher positional accuracy, while regions with higher values were flagged as less reliable, typically corresponding to flexible or disordered regions. 

Structural refinement and analysis: The selected models were further evaluated for structural consistency using tools such as Discovery Studio. These assessments included the identification of secondary structure elements, hydrogen bond networks, and potential disulfide bridges. Where necessary, the models were energy-minimized to resolve steric clashes and ensure conformational stability before further analysis.

### 4.4. Molecular Docking

The 3D structures of the Kv1.3 channel protein and ShK domain peptides were prepared and refined using the Prepare Protein tool in Discovery Studio 2021 [58]. Missing side chains and residues were reconstructed and optimized, while disulfide bonds were assigned automatically. Hydrogen atoms were added, and protonation states of titratable residues were adjusted to pH 7.4. Solvent-exposed loops and non-crystallographic artifacts were refined using the Clean Protein Surface tool. The larger protein was designated as the receptor, while the smaller or interacting protein was assigned as the ligand.

Protein–protein docking was performed using ZDOCK, a Fast Fourier Transform-based rigid-body docking algorithm. Parameter configuration: Angular sampling was set to 6 degrees to generate a comprehensive set of docking poses. The default ZDOCK scoring function, which combines shape complementarity, desolvation energy, and electrostatic interactions, was applied. The top 2000 docking poses were retained for subsequent analysis.

Post-docking analysis involved re-ranking the top 10 poses from ZDOCK using the RDOCK module, which refines docking poses through CHARMm energy calculations, incorporating van der Waals and electrostatic interaction terms. The pose with the lowest interaction energy was selected as the most favorable protein-peptide complex.

Interaction analysis was performed using the Analyze Protein–Protein Interactions tool. This analysis identified critical interactions, including hydrogen bonds, salt bridges, hydrophobic contacts, and potential adverse interactions, providing insights into the binding interface.

### 4.5. Molecular Dynamics Simulation

Molecular dynamics simulations were performed to study the protein-ligand complex using Discovery Studio 2021. The protein structure was prepared by reconstructing missing residues and side chains, assigning disulfide bonds, and optimizing the structure with the Prepare Protein module. The ligand was sketched, optimized, and protonated at physiological pH (7.4) using the Prepare Ligands and Ligand Ionization tools. The protein-ligand complex was assembled, protonated according to the CHARMm force field, and solvated in a cubic periodic box of Transferable Intermolecular Potential Three-Point (TIP3P) water molecules with a 10 Å margin. Counterions were added to neutralize the system and ensure proper electrostatic balance.

Energy minimization was conducted in two stages: steepest descent (5000 steps) followed by conjugate gradient minimization (10,000 steps or until convergence). The system was equilibrated using an NVT ensemble (300 K, 100 ps) followed by an NPT ensemble (1 atm, 500 ps) to stabilize density. A 50 ns production simulation was run in the NPT ensemble at 300 K with a 1 fs time step, applying the SHAKE algorithm for hydrogen bond constraints and Particle Mesh Ewald for long-range electrostatics.

Post-simulation analysis included RMSD and RMSF calculations to assess structural stability and flexibility, hydrogen bond analysis to evaluate interactions between the protein and ligand, and MM-PBSA calculations to estimate binding energy.

## 5. Conclusions

This study utilized AlphaFold-driven molecular modeling, combined with docking and molecular dynamics simulations, to systematically analyze the binding modes and mechanisms of ShK domain peptides from four species of sea anemone in the South China Sea with the Kv1.3 channel. ShK domain peptides from different sea anemone species exhibited distinct characteristics in amino acid composition, loop length, functional residue distribution, charge properties, target sites, and binding mechanisms. Peptides derived from *Heteractis magnifica*, *Heteractis crista*, and *Exaiptasia Diaphana* mainly target the pore domain of the Kv1.3 channel, functioning as traditional pore-blocking toxins. In contrast, peptides derived from *Macrodactyla doreensis* exhibited unique features in hydrophobic interactions and the adaptability of bulky residues, predominantly targeting the voltage-sensing domain of the Kv1.3 channel as gating modifiers. This study highlights the potential of ShK domain peptides in modulating the Kv1.3 channel and introduces an effective approach for large-scale prediction and evaluation of the structural and functional properties of peptides.

## Figures and Tables

**Figure 1 marinedrugs-23-00085-f001:**
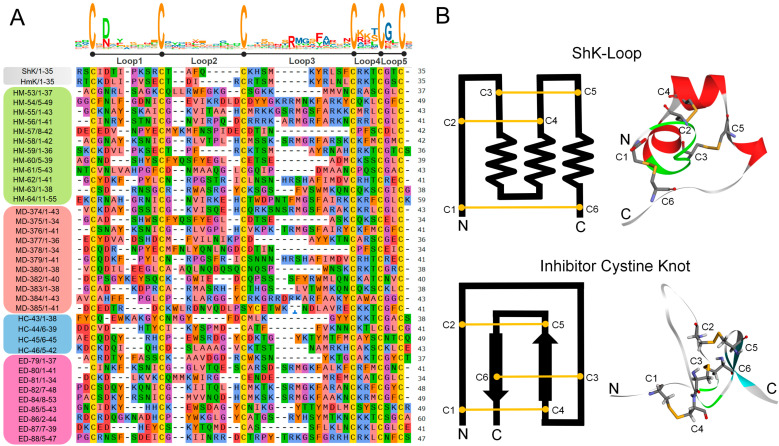
Sequence analysis of 36 ShK domain peptides from four species of sea anemone in the South China Sea. (**A**) Multiple sequence alignment of ShK domain peptides in this study. Residues are shown by physicochemical properties following Zappo colors. The reference ShK domain peptides shown are ShK (Uniport: P29187) from the sea anemone *Stichodactyla helianthus*; HmK (Uniport: O16846) from the Magnificent sea anemone *Heteractis magnifica*. (**B**) Typical structural features of ShK domain peptide and inhibitor cystine knot-like peptide illustrate the arrangement and connectivity of its disulfide bonds. Note that in the multiple sequence alignment, only the two amino acids preceding the first cysteine and the two amino acids following the last cysteine were retained in these peptides. Loop represents a cysteine-constrained amino acid sequence. The HM, HC, ED, and MD series peptides are derived from *Heteractis magnifica, Heteractis crispa, Exaiptasia diaphana,* and *Macrodactyla doreensis*, respectively. The sequence logo represents the conserved residues within each loop, where the height of each letter indicates its relative frequency at that position.

**Figure 2 marinedrugs-23-00085-f002:**
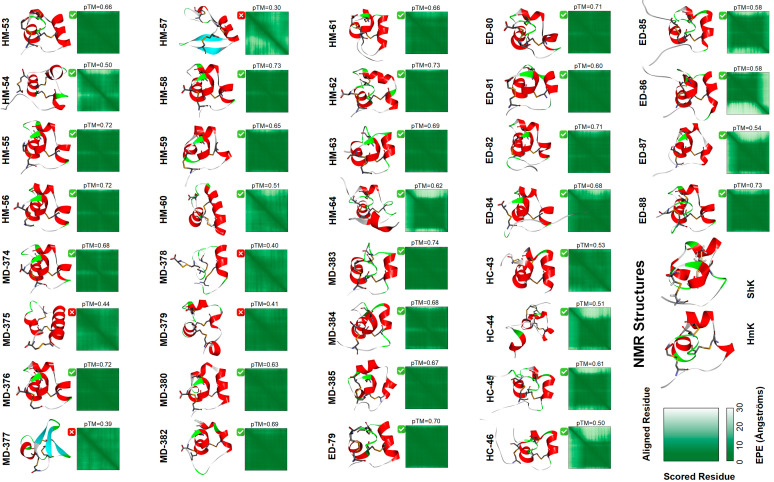
The model structures of the 36 ShK domain peptides investigated in this study predicted by AlphaFold3 and the NMR-determined structures of ShK (PDB: 1ROO) and HmK (PDB: 6EI7). The disulfide bonds are highlighted in yellow. The HM, HC, ED, and MD series peptides are derived from *Heteractis magnifica*, *Heteractis crispa*, *Exaiptasia diaphana*, and *Macrodactyla doreensis*, respectively.

**Figure 3 marinedrugs-23-00085-f003:**
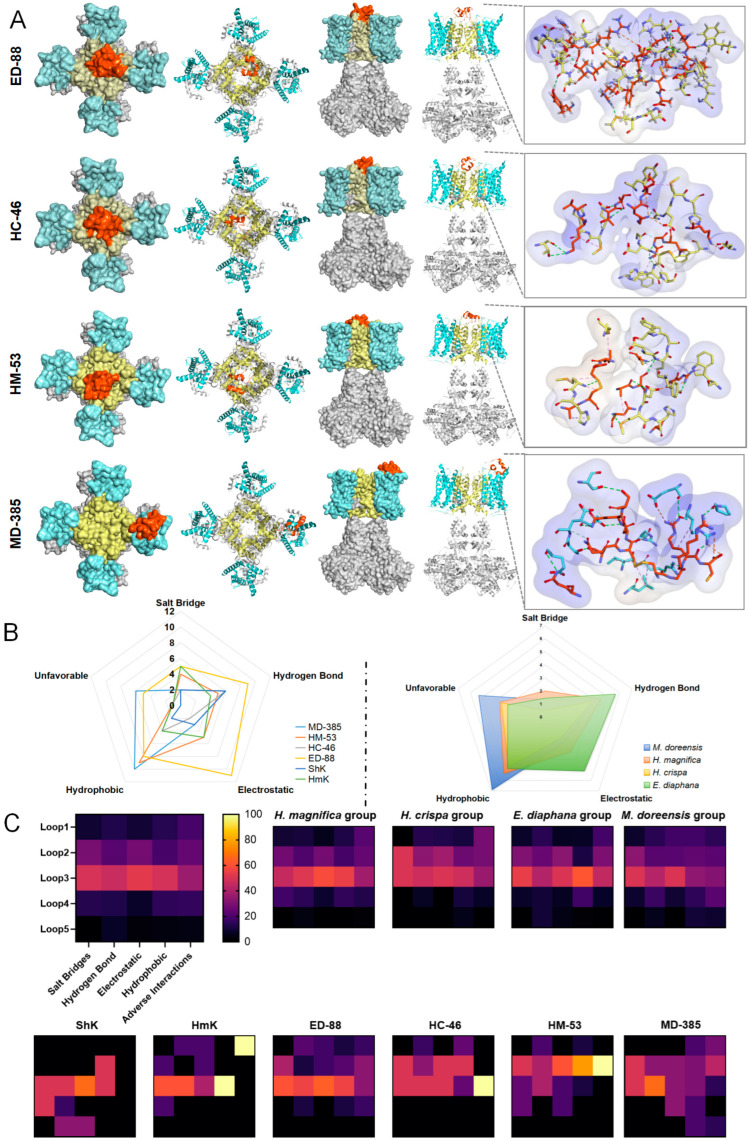
Binding interaction analysis of ShK domain peptides with the Kv1.3 channel. (**A**) Molecular docking analyses illustrating the binding interactions between the top-ranked peptides from each group (colored orange) and the Kv1.3 channel (colored grey) (PDB: 7EJ1). (**B**) Interaction number distributions between the top-ranked peptides from four groups, as well as reference peptides, with the Kv1.3 channel. The average number of interactions for each group of peptides is also shown. (**C**) Distribution of interaction sites across Loop1–Loop5 of the top-ranked peptides and reference peptides. The average distribution of interaction sites across all four groups is provided for comparison.

**Figure 4 marinedrugs-23-00085-f004:**
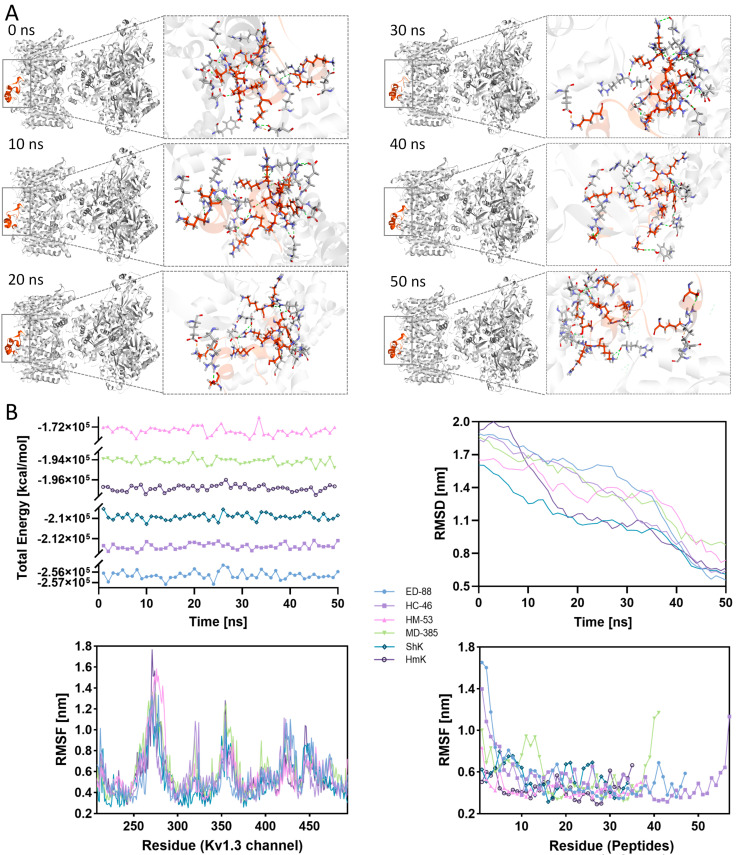
Molecular dynamic simulation and binding interaction analysis of ShK domain peptides. (**A**) Molecular dynamic simulation analyses showing the binding interaction between ED-88 (colored orange) and Kv1.3 channel (colored grey). (**B**) Molecular dynamics simulation data analysis for the top-ranked peptides and the reference peptides, including free binding energy (top left), root mean square deviation (top right), root mean square fluctuation of the channel protein in the complex (bottom left), and root mean square fluctuation of the peptide in the complex (bottom right).

**Figure 5 marinedrugs-23-00085-f005:**
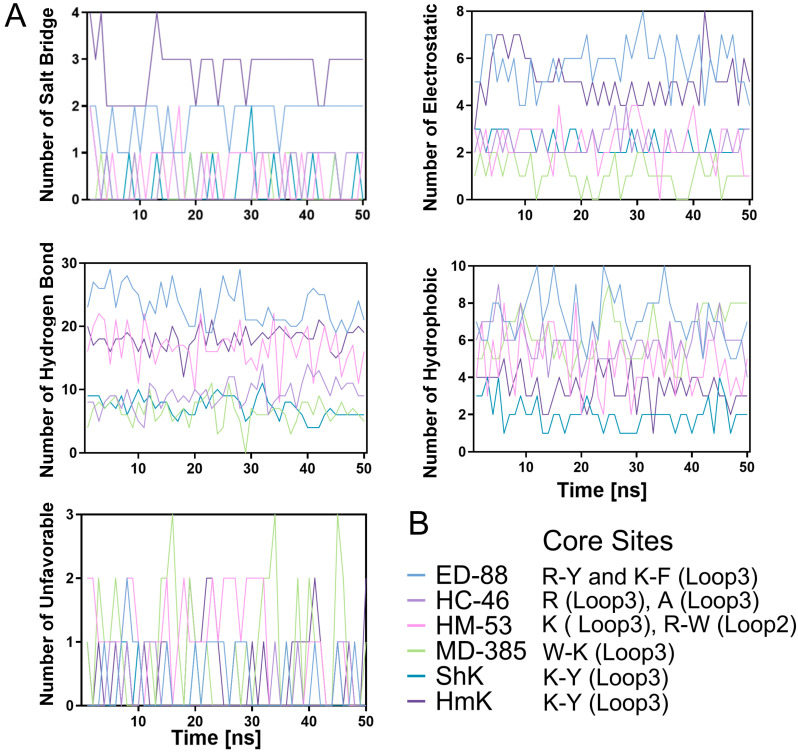
Core binding sites and interactions of ShK domain peptides with the Kv1.3 channel. (**A**) Molecular dynamics simulations showing the binding interaction between top-ranked peptides and the reference peptides with the Kv1.3 channel. (**B**) Core binding sites of top-ranked peptides and reference peptides.

## Data Availability

The data that support the findings of this study are available from the corresponding author, G, upon reasonable request.

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
