# Peer review of "Large-Scale AI-Based Structure and Activity Prediction Analysis of ShK Domain Peptides from Sea Anemones in the South China Sea"

_marinedrugs, 2025, doi:10.3390/md23020085_

Round 1
Reviewer 1 Report
Comments and Suggestions for Authors
The authors present an interesting in silico analysis of 36 amino acid sequences corresponding to peptides with ShK domains. These sequences were predicted from transcriptomic data previously generated from four species of sea anemones (Heteractis magnifica, Heteractis crispa, Exaiptasia diaphana, and Macrodactyla doreensis) collected from the waters of the South China Sea. The sequences were analyzed by comparing them with ShK and HmK toxins through multiple sequence alignment. Subsequently, structural models of the 36 sequences were obtained using AlphaFold3, followed by molecular docking between the Kv1.3 channel and the 36 putative peptides containing ShK domains. Finally, the peptide-channel interaction was simulated using molecular dynamics.
Their results showed that ShK domain-containing peptides from different sea anemone species exhibited distinctive features in amino acid composition, loop length, functional residue distribution, charge properties, target sites, and binding mechanisms.
Interestingly, they identified that peptides derived from Macrodactyla doreensis exhibited differences in their interaction mode compared to the other peptides, which followed a typical pore-blocking toxin mechanism. The prediction suggests that peptides from M. doreensis, particularly ED-88, act on the Kv1.3 channel as gating modifiers. This study provides an interesting framework for the rational selection of candidates obtained through high-throughput sequencing for further experimental characterization.
Comments:
I find the study interesting and well-designed in terms of in silico experimentation. However, I would have liked to see an evaluation of the recombinant toxins' activity on channels expressed in cells—at least for two of them, considering ED-88 and a candidate from the group expected to act as blocking toxins. This would provide in vivo verification of their in silico conclusions. In this regard, while in vivo or in vitro validation of activity and structure, respectively, is not yet available, I recommend that the authors use more measured language in some statements. For example, in lines 541 and 542, I suggest rewording the sentences to indicate that the results strongly suggest rather than definitively confirm the conclusions. As the authors themselves acknowledge, these methods still have limitations.
As a final suggestion, I think it would be helpful to include a flowchart illustrating the steps and tools used in this study. This would enhance clarity and allow readers to better understand the methodology at a glance.
I believe the study is suitable for publication, as its in silico analysis is robust and provides valuable insights.
Author Response
Responses to the reviewer 1’s comments:
- I find the study interesting and well-designed in terms of in silico experimentation. However, I would have liked to see an evaluation of the recombinant toxins' activity on channels expressed in cells—at least for two of them, considering ED-88 and a candidate from the group expected to act as blocking toxins. This would provide in vivo verification of their in silico conclusions.
Response: We greatly appreciate your valuable comment. Due to the presence of three disulfide bonds in each of these peptides, site-specific oxidation is challenging, resulting in a relatively low yield. Therefore, we are actively exploring strategies to optimize the synthesis and oxidative folding efficiency. We expect to perform solid-phase synthesis of several peptides (including ED-88), which performed well in our predictions, in approximately six months. These linear peptides will then undergo oxidative folding to form peptides with specific spatial structures, which will be used for activity testing and target screening. Unfortunately, obtaining a sufficient quantity of these peptides for functional tests will take approximately six months, even under optimal conditions. As a result, functional validation is beyond the scope of the present study but will be a key focus in our next phase of investigating these promising peptides.
- In this regard, while in vivo or in vitro validation of activity and structure, respectively, is not yet available, I recommend that the authors use more measured language in some statements. For example, in lines 541 and 542, I suggest rewording the sentences to indicate that the results strongly suggest rather than definitively confirm the conclusions. As the authors themselves acknowledge, these methods still have limitations.
Response: Thank you very much for this constructive comment. We have revised "After binding with the peptides, the Kv1.3 channel showed a reduction in local fluctuations, highlighting the exceptional ability of these peptides to restrict local channel dynamics" to "Upon binding with these peptides, the Kv1.3 channel exhibits a reduction in local fluctuations, suggesting that these peptides may possess a unique ability to regulate local channel dynamics (Page 15, Line 556-558)."
- As a final suggestion, I think it would be helpful to include a flowchart illustrating the steps and tools used in this study. This would enhance clarity and allow readers to better understand the methodology at a glance.
Response: Thank you very much for your valuable feedback, which is very enlightening for us. However, we prefer to use concise subtitles in the “Materials and Methods” section, i.e. Alignment→ Phylogenetic analyses → AlphaFold3 Modeling → Molecular docking → Molecular dynamics simulation, to clearly illustrate the logical flow of the experimental design. Detailed descriptions of each step and the tools used, which are hard to illustrate in a flowchart, are provided in the text to ensure clarity and reproducibility.
Reviewer 2 Report
Comments and Suggestions for Authors
The authors tried to analyze and search for potential ShK domain peptides within 36 peptides that derived from four different Sea Anemones using combination of AlphaFold 3, molecular docking and MD. Overall, the paper is well-structured and well-written. However, there are some concerns still need to be addressed.
Major concerns:
- Page 3, Line117-118, the authors mentioned “peptides from the ED group are characterized by an abundance of acidic residues, especially in Loop2 and Loop3”. I calculated the sum of Glu and Asp in Loop2 and Lopp3 in each group and found that HM (17) and MD (17) groups also showed relatively higher distribution of both them compared to the two reference peptides (ED was 15). So, it needed to reorganize this sentence, especially the phrase “an abundance of” is not a precise expression for describing these results.
- Page 4, Line 161, the authors mentioned the lowest EPE score, but the related figure or table cannot be found in entire manuscript.
- Page 12, Line 374-375, based on Figure 4B about RMSF, the authors conclude that Kv1.3 channel showed significant fluctuations near residues R350 and M450. However, the biggest fluctuations came from residues within 260~270, so you should mention the explanations what is that means.
Miner comments:
- Page 3, the first paragraph is describing the MSA results for 36 peptides, so you need to add “Figure 1A” in a appropriate location. In the current version, this figure information is missing.
- Figure 2, the figure legend is too rough, you should clarify the cyan and green fluorescence represent what.
- Figure 4B, the size scale of the neighboring two graphs are so different, left one is significantly bigger than the right one including fonts.
Author Response
Responses to the reviewer 2’s comments:
- Page 3, Line117-118, the authors mentioned “peptides from the ED group are characterized by an abundance of acidic residues, especially in Loop2 and Loop3”. I calculated the sum of Glu and Asp in Loop2 and Lopp3 in each group and found that HM (17) and MD (17) groups also showed relatively higher distribution of both them compared to the two reference peptides (ED was 15). So, it needed to reorganize this sentence, especially the phrase “an abundance of” is not a precise expression for describing these results.
Response: Thanks for your helpful comment. Following your suggestion, we recalculated the data. The total number of peptides in the HM, MD, HC and ED groups is 12, 11, 4 and 9, respectively. The total number of acidic amino acids in each group is 17, 17, 15, and 5, respectively. The ratios of acidic amino acids to total peptides are 1.42, 1.55, 1.67, and 1.25, compared to ShK and HmK, which have ratios of 0 and 1, respectively. Accordingly, we have reorganized and revised the description accordingly. The previous expression in this section was indeed incorrect, so we have revised the statement: "Peptides from the E. diaphana group are characterized by an abundance of acidic residues, especially in Loop2 and Loop3. These peptides deviate from the classical distribution seen in the reference sequences, with substitutions introducing Glutamate Acid (Glu) or Aspartic acid (Asp)" to "Relative to the reference sequence, all peptide groups introduced a higher proportion of acidic amino acids in Loop2 and Loop3, leading to a deviation from the canonical distribution, with the E. diaphana group showing the highest proportion (Page 3, Line 123-125)." Thanks for your helpful comment.
- Page 4, Line 161, the authors mentioned the lowest EPE score, but the related figure or table cannot be found in entire manuscript.
Response: Thanks for your helpful comment. We sincerely apologize for not clearly and distinctly displaying the EPE score in Figure 2. To address this, we have modified the figure legend to make them more prominent. Additionally, we have referred to 'Figure 2' in this section (Page 4, Line 175).
- Page 12, Line 374-375, based on Figure 4B about RMSF, the authors conclude that Kv1.3 channel showed significant fluctuations near residues R350 and M450. However, the biggest fluctuations came from residues within 260~270, so you should mention the explanations what is that means.
Response: Thanks for your constructive comments. We have provided an explanation for this phenomenon. This is due to the presence of an unstructured coil near residue A265, located at the periphery of the Kv1.3 channel, which lacks a defined secondary structure. This leads to significant structural fluctuations, and correspondingly, a high RMSF value (Page 12, Line 396-399).
- Page 3, the first paragraph is describing the MSA results for 36 peptides, so you need to add “Figure 1A” in a appropriate location. In the current version, this figure information is missing.
Response: Thanks for your constructive comments. We have referred to Figure 1A in the first line of the first paragraph (Page 3, Line 110).
- Figure 2, the figure legend is too rough, you should clarify the cyan and green fluorescence represent what.
Response: Thanks for your helpful comment. We have revised the figure legend in Figure 2 to make it clearer and more explicit (Figure 2).
- Figure 4B, the size scale of the neighboring two graphs are so different, left one is significantly bigger than the right one including fonts.
Response: Thanks for your helpful comment. We have realigned the graphs and make the font size consistent (Figure 4B).
Reviewer 3 Report
Comments and Suggestions for Authors
Hua et al. investigate the binding modes and mechanisms with the Kv1.3 channel of 36 ShK domain peptides obtained from four species (H. magnifica, H. crispa, E. diaphana and M. doreensis) of sea anemone in the South China Sea. This study was performed by using AlphaFold-driven molecular modeling, docking and molecular dynamics simulations. As a result, it was shown that their ShK domain peptides are distinct in terms of amino acid composition, loop length, functional residue distribution, charge properties, target sites and binding mechanisms and that the stability and specificity of the binding of the ShK domain peptides and Kv1.3 channel are affected by variations among the ShK domain peptides in loop length, residue composition and charge distribution. It was concluded that the ShK domain peptides have an ability to modulate the Kv1.3 channel. Although this manuscript is well written, some simple mistakes are noticed. There are several points that may serve to improve this manuscript so that it is easier for non-specialists to understand, as follows:
1. Line 45: it will be better to shortly explain “cystine-knot motif”.
2. Line 51: not “particularly” but “, particularly”?
3. Line 55: it will be better to shortly explain “CCR7-TEM”.
4. Line 63: not “actie” but “active”?
5. Line 64: not “psoriasisexamplifing” but “psoriasis examplifing”.
6. Lines 94 and 95: the four species names should be in italic (see page 3).
7. Lines 104 and 130: the writing style of these subtitles differs. Please amend this point.
8. Line 120: not “glutamate” but “glutamic acid”?
9. Line 136: it is unknown what “pTM” means without reference to 598 and 599 lines that follow. Please amend this point.
10. Fig. 1: it is difficult to clearly read the text above “loop1, loop2 ..”. Please amend this point. In the legend of Fig. 1, it should be mentioned what “HM”, “MD”, “HC” and “ED” written in Fig. 1 are. This is true for “MSA” (see line 105). It is unknown what “ICK” means without reference to line 471. Please amend this point. It may be better to give a list of the abbreviations used in this manuscript. Not “Heteractis magnifica” but “H. magnifica” (see line 94).
11. Lines 161 and 162: it is not necessary to repeatedly define “EPE” (see lines 147 and 148).
12. Lines 176 and 177: not “.. contribute distinct properties ..” but “.. contribute to distinct properties ..”?
13. Fig. 2: in the legend of Fig. 2, it should be mentioned what “HM”, “MD”, “HC” and “ED” written in Fig. 2 are. If possible, it might be better if the green in the square was more distinct. Not “NMR Struction” but “NMR Structure”?
14. Lines 210 and 212: it is not necessary to repeatedly define “PD” and “VSD” (see line 49). As stated in the above comment 8, it may be better to give a list of the abbreviations used in this manuscript.
15. Fig. 3A: if possible, it might be better if this figure was more distinct.
16. Line 332: not “R residues” but “Arg residues”?
17. Line 335: not “exhibithigher” but “exhibit higher”.
18. Line 345: it will be better to shortly explain “CHARMm”.
19. Fig. 4A: if possible, it might be better if this figure was more distinct.
20. The legend of Fig. 4: “RMSD” and “RMSF” should be spelled out, although this was done in the text.
21. Line 439: not “and” but “and”.
22. Line 465: not “The” but “the”.
23. Line 597: not “Among” but “among”.
24. Lines 625 and 638: pleas use either “CHARMm” or “CHARMM”.
25. Line 633: “Molecular dynamics” does not seem to be defined as “MD”.
26. Line 639: it will be better to shortly explain “TIP3P”.
27. References: there is no consistency in the way the references are written; for example, compared the titles of Refs. 4 and 5 and also the titles of Refs. 44 and 45. There also seems to be co consistency in the way Journal names are written. There are two Journal names in Ref. 11. These points should be amended.
There may be more mistakes than pointed out above. This manuscript should be checked very carefully.
Author Response
Responses to the reviewer 3’s comments:
- Line 45: it will be better to shortly explain “cystine-knot motif”.
Response: We sincerely apologize for the oversight. This is not a cysteine-knot, but rather a loop structure corresponding to the cysteine-knot. We have revised to clarify this point changing "This highly conserved cystine-knot motif provides critical support for its stable three-dimensional structure and biological activity" to "This specific disulfide bond connection enables these peptides to form a relatively rigid three-dimensional structure, similar to a loop structure, providing crucial support for their stable three-dimensional structure and biological activity" (Page 2, Line 46-49).
- 2. Line 51: not “particularly” but “, particularly”?
Response: We have reorganized the sentence and revised "The Kv1.3 channel is widely expressed in the immune and nervous systems of mammals, playing a pivotal role in the activation and differentiation of T lymphocytes and various immune cells" to "The Kv1.3 channel is widely expressed in the immune and nervous systems of mammals, where it plays a pivotal role, particularly in the activation and differentiation of T lymphocytes and various immune cells" (Page 2, Line 52-55).
- Line 55: it will be better to shortly explain “CCR7-TEM”.
Response: We have now provided the full name for CCR7-TEM, which is CCR7-effector memory T lymphocytes (Page 2, Line 58-59).
- Line 63: not “actie” but “c”?
Response: We have revised "actie" to "active" (Page 2, Line 68).
- Line 64: not “psoriasisexamplifing” but “psoriasis examplifing”.
Response: We have revised "psoriasisexamplifing" to "psoriasis examplifing" (Page 2, Line 68).
- Lines 94 and 95: the four species names should be in italic (see page 3).
Response: We have now made all the species names italic (Page 2-3, Line 99-100).
- Lines 104 and 130: the writing style of these subtitles differs. Please amend this point.
Response: We have now made the writing styles consistent (Page 3, Line 110-128).
- Line 120: not “glutamate” but “glutamic acid”?
Response: We have reorganized the sentence and revised "Peptides from the E. diaphana group are characterized by an abundance of acidic residues, especially in Loop2 and Loop3. These peptides deviate from the classical distribution seen in the reference sequences, with substitutions introducing Glutamate (Glu) or Aspartic acid (Asp)" to "Relative to the reference sequence, all peptide groups introduced a higher proportion of acidic amino acids in Loop2 and Loop3, leading to a deviation from the canonical distribution, with the E. diaphana group showing the highest proportion" (Page 3, Line 123-125).
- Line 136: it is unknown what “pTM” means without reference to 598 and 599 lines that follow. Please amend this point.
Response: We have added an introduction to the pTM, which is predicted TM-score (Page 3, Line 144).
- Fig. 1: it is difficult to clearly read the text above “loop1, loop2 ..”. Please amend this point. In the legend of Fig. 1, it should be mentioned what “HM”, “MD”, “HC” and “ED” written in Fig. 1 are. This is true for “MSA” (see line 105). It is unknown what “ICK” means without reference to line 471. Please amend this point. It may be better to give a list of the abbreviations used in this manuscript. Not “Heteractis magnifica” but “H. magnifica” (see line 94).
Response: We have annotated the Loop and "HM", "MD", "HC" and "ED" in the Figure 1 legend for better understanding. The full names of MSA and ICK have now been provided in the Fig 1 legend. In addition, we added all abbreviations at the end of the article, based on your suggestion. In addition, we have added annotated of the sequence logo, which is "sequence logo represent the conserved residues within each loop, where the height of each letter indicates its relative frequency at that position" (Page 4, Line 159-170).
- Lines 161 and 162: it is not necessary to repeatedly define “EPE” (see lines 147 and 148).
Response: We have revised "Expected Position Error (EPE)" to "EPE " (Page 4, Line 174).
- Lines 176 and 177: not “.. contribute distinct properties ..” but “.. contribute to distinct properties ..”?
Response: We have revised "contribute distinct properties" to "contribute to distinct properties" (Page 5, Line 191).
- Fig. 2: in the legend of Fig. 2, it should be mentioned what “HM”, “MD”, “HC” and “ED” written in Fig. 2 are. If possible, it might be better if the green in the square was more distinct. Not “NMR Struction” but “NMR Structure”?
Response: We have added explanations for "HM", "MD", "HC" and "ED" in the annotations of Figure 2 and the NMR Structure has been corrected to "NMR Structure". In addition, we have fine-tuned the green color in the square block to make it clearer (Figure 2).
- Lines 210 and 212: it is not necessary to repeatedly define “PD” and “VSD” (see line 49). As stated in the above comment 8, it may be better to give a list of the abbreviations used in this manuscript.
Response: We have revised "pore domain (PD)" to " PD" and "voltage-sensing domain" to " VSD" (Page 6, Line 226, 228).
- Fig. 3A: if possible, it might be better if this figure was more distinct.
Response: The number of non-bond changes here has been shown in the chart of Figure 3B.
- Line 332: not “R residues” but “Arg residues”?
Response: We have revised "positively charged R residues" to "positively charged Arg residues" (Page 10, Line 349).
- Line 335: not “exhibithigher” but “exhibit higher”.
Response: We have revised "Peptides from the M. doreensis group exhibithigher levels of unfavorable hydrophobic contacts, largely due to their low sequence conservation" to "Peptides from the M. doreensis group exhibit higher levels of unfavorable hydrophobic contacts, largely due to their low sequence conservation" (Page 10, Line 352).
- Line 345: it will be better to shortly explain “CHARMm”.
Response: We have added an introduction to the Chemistry at Harvard Macromolecular Mechanics (CHARMm) that it is a molecular force field for molecular dynamics (Page 10, Line 362-363).
- Fig. 4A: if possible, it might be better if this figure was more distinct.
Response: The number of hydrogen bond changes here has been shown in the chart of Figure 5.
- The legend of Fig. 4: “RMSD” and “RMSF” should be spelled out, although this was done in the text.
Response: We have revised "RMSD" to "root mean square deviation" and “RMSF” to “root mean square fluctuation” (Page 11, Line 373-374).
- Line 439: not “and” but “and”.
Response: We have revised "H. magnifica, H. crispa, E. diaphana, and M. doreensis" to "H. magnifica, H. crispa, E. diaphana, and M. doreensis " (Page 14, Line 464).
- Line 465: not “The” but “the”.
Response: We have revised "Notably, The majority of peptides retained the critical Lys-Phe or Tyr or Trp functional dyad in Loop3" to "Notably, the majority of peptides retained the critical Lys-Phe or Tyr or Trp functional dyad in Loop3" (Page 14, Line 490).
- Line 597: not “Among” but “among”.
Response: We have revised "Model selection, Among the five structural models generated by AlphaFold for each protein, the top-ranked model was selected based on the pTM score" to "Model selection, among the five structural models generated by AlphaFold for each protein, the top-ranked model was selected based on the pTM score" (Page 17, Line 625).
- Lines 625 and 638: pleas use either “CHARMm” or “CHARMM”.
Response: We have revised "protonated according to the CHARMM force field" to "protonated according to the CHARMm force field" (Page 18, Line 668).
- Line 633: “Molecular dynamics” does not seem to be defined as “MD”.
Response: We have revised "MD simulations were performed to study the protein-ligand complex using Discovery Studio 2021" to "Molecular dynamics simulations were performed to study the protein-ligand complex using Discovery Studio 2021" (Page 18, Line 662).
- Line 639: it will be better to shortly explain “TIP3P”.
Response: We have added the full name for TIP3P, i.e. Transferable Intermolecular Potential 3-Point (TIP3P) (Page 18, Line 669).
- References: there is no consistency in the way the references are written; for example, compared the titles of Refs. 4 and 5 and also the titles of Refs. 44 and 45. There also seems to be co consistency in the way Journal names are written. There are two Journal names in Ref. 11. These points should be amended.
Response: We have double checked the references and made the reference style consistent (Page 19, Line 723-728, 741-742; Page 21, Line 822-827).
In addition, we carefully examined this manuscript and found some capitalization errors and unnecessary spaces, which were corrected (Page 2, Line 96; Page 5, Line 188, 208; Page 8, Line 274; Page 12, Line 374, 376). Deleted some unnecessary abbreviations (Page 1, Line 39-40; Page 2, Line 62; Page 9, Line 317; Page 13, Line 442; Page 16, Line 601, 605; Page 17, Line 623, 646; Page 18, Line 674).